# Management of Dyspepsia and Gastroparesis in Patients with Diabetes. A Clinical Point of View in the Year 2021

**DOI:** 10.3390/jcm10061313

**Published:** 2021-03-23

**Authors:** Silvia Bonetto, Gabriella Gruden, Guglielmo Beccuti, Arianna Ferro, Giorgio Maria Saracco, Rinaldo Pellicano

**Affiliations:** 1Unit of Gastroenterology, Molinette Hospital, 10126 Turin, Italy; giorgiomaria.saracco@unito.it (G.M.S.); rinaldo_pellican@hotmail.com (R.P.); 2Department of Medical Sciences, University of Turin, 10126 Turin, Italy; gabriella.gruden@unito.it (G.G.); guglielmo.beccuti@unito.it (G.B.); arianna.ferro@unito.it (A.F.)

**Keywords:** dyspepsia, diabetes, gastroparesis

## Abstract

Diabetes mellitus is a widespread disease, and represents an important public health burden worldwide. Together with cardiovascular, renal and neurological complications, many patients with diabetes present with gastrointestinal symptoms, which configure the so-called diabetic enteropathy. In this review, we will focus on upper gastrointestinal symptoms in patients with diabetes, with particular attention to dyspepsia and diabetic gastroparesis (DG). These two clinical entities share similar pathogenetic mechanisms, which include autonomic neuropathy, alterations in enteric nervous system and histological abnormalities, such as interstitial cells of Cajal depletion. Moreover, the differential diagnosis may be challenging because of overlapping clinical features. Delayed gastric emptying should be documented to differentiate between DG and dyspepsia and it can be assessed through radioactive or non-radioactive methods. The clinical management of dyspepsia includes a wide range of different approaches, above all *Helicobacter pylori* test and treat. As regards DG treatment, a central role is played by dietary modification and glucose control and the first-line pharmacological therapy is represented by the use of prokinetics. A minority of patients with DG refractory to medical treatment may require more invasive therapeutic approaches, including supplemental nutrition, gastric electric stimulation, pyloromyotomy and gastrectomy.

## 1. Introduction

Diabetes mellitus (DM) is a widespread disease. According to the last estimate from the International Diabetes Federation (ID) it affects 463 million people worldwide with increasing prevalence [1]. DM represents an important public health burden, mainly because of its cardiovascular, renal and neurological complications. In addition, many patients with diabetes present with upper gastrointestinal (GI) symptoms and motility alterations. Among the latter, delayed gastric emptying (GE) affects up to 50% of patients with both type 1 and type 2 DM manifesting with dyspepsia, gastroparesis or, for a proportion of patients, remaining asymptomatic [2]. As dyspepsia and diabetic gastroparesis (DG) share similar pathogenetic mechanisms and clinical features, the differential diagnosis may be challenging. Recently, some authors suggested that functional dyspepsia (FD) and DG could be different expressions of the same spectrum of gastric neuromuscular disorders, with common histopathological alterations and comparable clinical manifestations and prognosis [3].

In this review, we will discuss an update of dyspepsia and gastroparesis in patients with diabetes, focusing on pathophysiology, clinical presentation and management of these manifestations.

## 2. Dyspepsia: Definition and Clinical Classification

The term dyspepsia includes a set of symptoms with epigastric localization, which can be episodic or persistent, with variable intensity and severity. In the clinical setting, it is often difficult to characterize these symptoms and to distinguish dyspepsia from other GI disorders such as gastroesophageal reflux disease (GERD) [4]. The American College of Gastroenterology (ACG) and Canadian Association of Gastroenterology (CAG) clinical guidelines give a useful definition of dyspepsia as predominant epigastric pain which lasts at least one month and is associated with any other upper GI symptom such as epigastric fullness, nausea, vomiting or heartburn [5].

For the appropriate clinical management, it is important to distinguish organic dyspepsia from FD. The former includes patients in whom clinical evaluation, laboratory tests, endoscopy or radiologic studies can identify a pathologic process which is the cause of dyspeptic symptoms, while FD includes all cases of dyspepsia without evidence of an organic cause [6]. The exclusion of organic causes requires endoscopy and, where needed, radiologic investigations, such as ultrasound or computed tomography, along with Helicobacter pylori (H. pylori) testing and treating and re-evaluation of symptoms after its eradication [7].

Functional dyspepsia can be classified on the basis of prevalent symptoms in postprandial distress syndrome (PDS) and epigastric pain syndrome (EPS) [8]. These two entities, however, have blurred boundaries as they frequently overlap and they share similar therapeutic strategies. Moreover, due to their common motility alterations, PDS is more likely to overlap with gastroparesis.

Although different definitions of FD have been previously proposed, the most recent update is represented by Rome IV criteria, shown in Table 1 [9].

*H. pylori*-associated dyspepsia represents a distinct form of dyspepsia [10]. If dyspepsia resolves six months after bacterial eradication it can be attributed to *H. pylori* infection [11,12] otherwise the disorder is deemed FD [7].

### 2.1. Organic Dyspepsia

The most common cause of organic dyspepsia is peptic ulcer disease, which is often associated with either *H. pylori* infection or chronic use of non-steroidal anti-inflammatory drugs (NSAIDs) [6]. *H. pylori* is a Gram-negative, micro-aerophilic bacterium, usually acquired during childhood, whose natural habitat is the luminal surface of the gastric epithelium [13]. Since the human stomach is an unfriendly place for microbial survival, *H. pylori* has developed a repertoire of acid resistance mechanisms which allow the microorganism to overcome the mucous layer. In particular, via the enzyme urease, the bacterium creates a cloud of acid neutralizing chemicals around it, offering protection from the acid [14]. *H. pylori* infection is accepted as the most important cause of gastritis and PUD in humans. Moreover, it is recognized as a risk factor for gastric cancer [15] along with potential involvement in the pathogenesis of several extra-gastric manifestations, ranging from hematological diseases (such as idiopathic thrombocytopenic purpura, iron deficiency anemia), to neurological diseases (for example, Parkinson’s disease and other forms of neurodegeneration and dementia) [16,17,18,19].

Gastric or esophageal cancers are less frequent causes of dyspepsia. However, malignancy has an important impact on prognosis and clinical management of affected patients and should be ruled out in those aged 60 or over or with other risk factors [5]. Neoplastic risk is increased in patients with *H. pylori* infection, family history of gastric malignancy, previous gastric surgery, immigrants from endemic areas, smokers, patients with high alcohol consumption or with a long history of heartburn [6]. Moreover, the risk of gastric cancer is nearly doubled in males [5]. Pancreatic diseases, such as acute and chronic pancreatitis, can present with dyspepsia too. Pancreatic pain, however, is often more severe than epigastric pain related to dyspepsia, moreover chronic pancreatitis is usually associated with weight loss and other symptoms due to pancreatic insufficiency [6]. Other GI diseases associated with dyspepsia include gallstones, superior mesenteric artery syndrome, eosinophilic esophagitis, amyloidosis and lymphomas. The diseases that should be considered in the differential diagnosis of dyspepsia are summarized in Table 2.

### 2.2. Functional Dyspepsia

The etiology of FD remains unclear. It is considered a multifactorial disease, related to genetic, environmental, and socio-cultural factors [10,20]. The pathogenesis of FD involves different mechanisms, such as delayed GE [21], gastric accommodation impairment [22], hypersensitivity to gastric distention, altered chemosensitivity and altered duodenal sensitivity to acids and lipids [6,23]. Impaired intestinal permeability is involved in FD pathogenesis too and it is related to mucosal inflammation [23].

Intestinal physiological functions are modulated by GI endocrine mediators, such as ghrelin, cholecystokinin (CCK), glucagon-like peptide-1 (GLP-1), motilin and glucose-dependent insulinotropic peptide (GIP). These molecules influence mucosal immune system, permeability and enteric nervous system (ENS) through endocrine and paracrine mechanisms and might contribute to development of dyspepsia, even if more studies are needed for a better understanding of their role in FD [10].

An emerging factor in FD development is the role of microbial imbalance of GI tract. The term human microbiota is referred to the rich microbial community, consisting of more than 10^14^ microorganisms, that colonizes the GI tract [24]. The perturbation of this ecosystem as well as of mucosal integrity allows bacterial translocation and plays a key role in the development of GI and systemic diseases [25,26,27]. Molecules produced by microbiota components can modify intestinal motility, and, at the same time, motility influences microbiota composition. Moreover, bacterial mediators can act like neurotransmitters, thus interacting with ENS. Gut microbiota and its mediators also influence intestinal permeability [28]. They can modify the composition of the mucus layer and tight junctions through modulation of genetic expression [10]. Moreover, alteration of the mucosal immune system, inflammatory response and modification of gut microbiota after an acute gastroenteritis can predispose to development of FD [12]. A previous GI infection is a risk factor for irritable bowel syndrome (IBS) and FD development in about 10% of patients [29]. The role of *H. pylori* in the development of FD is complex and controversial. Whilst the bacterium induces alterations in gastric acid secretion, gut hormones production and motility [10], studies have reported conflicting results regarding changes to post-prandial gastric motility in *H. pylori*-infected patients [11].

As in other functional GI disorders, a central role in FD is played by the interaction between the GI tract and central nervous system. The main site of this two-way effect is amygdala, which is involved in emotions and pain and in satiety and fullness perception [10]. The brain-gut axis is structurally constituted by direct connections between the central nervous system and myenteric plexus. It is through this pathway that emotion can influence GE, intestinal motility, mucosal secretion and barrier function. Conversely, mental function can be influenced by GI motility, visceral inflammation and injury [8]. As a result of this interaction, mood disorders and psycho-social factors have a demonstrated relationship with both FD development and reduction in quality of life of these patients [23]. Therefore, FD and other functional GI disorders can be defined as the result of the interaction of biopsychosocial factors and gut physiology through the brain-gut axis.

## 3. Dyspepsia in Diabetic Patients

Dyspeptic symptoms are a frequent finding in patients with diabetes and they are part of the so-called diabetic enteropathy (DE), which includes the GI manifestations of DM [30]. Autonomic neuropathy has an important pathogenetic role in DE, together with interstitial cells of Cajal depletion and reduced expression of neuronal nitric oxide syntethase [30]. These alterations lead to abnormal GI motility, causing symptoms such as dyspepsia, nausea, vomiting, constipation and fecal incontinence.

Despite the high incidence of dyspepsia in patients with diabetes, the current literature offers limited data about the clinical features and the appropriate management of dyspepsia in this population. In case of a patient with DM presenting with upper GI symptoms, organic disease and medication side effects should be excluded: GLP-1 analogues, for example, can cause nausea and vomiting [30]. Moreover, DG should be excluded through GE measurement, as discussed below [2,31]. When organic dyspepsia, medication side effects and DG are excluded, the clinical management is analogous to that of non-diabetic patients, except for a more important therapeutic role of prokinetics in patients with DM.

## 4. *H. pylori* Infection in Patients with Diabetes

Many studies have analyzed the prevalence of *H. pylori* infection in symptomatic and asymptomatic patients with DM [32,33,34]. Hyperglycemia has been suggested as a predisposing factor for *H. pylori* colonization [32]. A recent case–control serological study demonstrated a significantly higher prevalence of *H. pylori* infection in patients with DM, who had positive antibody titers in 50.7% of cases, compared to 38.2% of controls [33]. Moreover, *H. pylori* positive patients showed higher incidence of GI symptoms, including bloating, distention, vomiting, abdominal pain, constipation and diarrhea, as well as systemic manifestations such as hypertension, muscular symptoms and chronic bronchitis, which is potentially attributable to *H. pylori* contribution to inducing systemic inflammation [33].

Among patients with DM, *H. pylori* infection has been shown to be higher in patients with gastroparesis, and bacterial eradication reduced symptoms such as upper abdominal pain and distention, early satiety and anorexia [34], thus suggesting a pathogenetic role of *H. pylori* in DG and reaffirming the therapeutic role of its eradication.

## 5. Diabetic Gastroparesis

DG is characterized by upper GI symptoms, such as epigastric distress, nausea, vomiting, early satiety or bloating, which occur in DM in the absence of organic obstruction. Epidemiologic studies about DG show heterogeneous data: in a study, among type 1 and type 2 diabetic patients with GI symptoms, the incidence of documented gastroparesis was 60% [35], while a more recent community-based study showed a ten-year cumulative incidence of gastroparesis of 5.2% in type 1 diabetes versus 1% in type 2 diabetes [36].

Whilst GE is often delayed in gastroparesis, the entity of motility alteration has a poor correlation with the severity of symptoms [37].

Glycemic control plays a key role in DG as it influences GE [2,38]. Severe acute hyperglycemia, in fact, has shown to delay GE in both healthy subjects and patients within type 1 DM, while its effects in type 2 diabetes are not clear [38,39]. Moreover, a prospective, observational, follow-up study showed that baseline levels of glycated hemoglobin (HbA1c) and duration of DM at baseline were independently associated with delayed GE, thus supporting a relationship between long-term hyperglycemic exposure and GE [40]. A subsequent cross-sectional study involving 147 type 2 diabetics confirmed the correlation of DG with blood glucose levels, HbA1c and duration of diabetes [41]. Currently, there are limited data on the long-term impact of improving glycemic control on patients with GE [38,42].

One of the main pathogenetic mechanisms of DG is autonomic neuropathy, characterized by loss of cells in motor and sensory sympathetic ganglia and structural changes of vagal nerve fibers, such as demyelination and axonal degeneration. These alterations often are multifocal, suggesting an ischemic injury [2]. Alterations in ENS and gut wall contribute to development of DG too and are a result of different processes, including apoptosis, oxidative stress, advanced glycation end products, and neuroimmune mechanisms [2]. Histological findings in both diabetic and non-diabetic gastroparetic patients showed loss of interstitial cells of Cajal (ICCs) and ganglion cells, fibrosis of the pylorus and lymphocytic infiltration around myenteric plexus [37]. Notably, gastroparesis and FD show the same histopathologichal changes, such as reduction of ICCs and anti-inflammatory C206+ macrophages, as demonstrated by histologic examination of full-thickness stomach biopsies [3]. These findings suggest a common pathophysiology and a possible target for new therapies, focused on the pathogenic mechanism of these diseases instead of mere symptom relief.

Comorbid abdominal pain with gastroparesis, has been related to visceral hypersensitivity, however, this symptom may be partly unrelated to gastric sensorimotor dysfunction. In a study of 32 patients with gastroparesis, 20 with comorbid DM, more than 60% had positive Carnett’s sign, which indicates somatic rather than visceral pain, and about half of them were hypervigilant to pain. Furthermore, more than one-third of these patients met criteria for neuropathic pain [43].

As in FD and other functional disorders [44], DG is associated with depression. In comparison with general population, diabetic patients have a higher prevalence of depression [45], which is often severe and has shown to play a role in expression of GI sensorimotor dysfunctions [2]. On the other side, DG has a negative impact on patients’ quality of life, with increased anxiety and depression [2].

In patients with DM and upper GI symptoms, gastroparesis can be diagnosed by the presence of delayed GE without gastric outlet obstruction [2]. The gold standard to define and quantify delayed GE is scintigraphy [46], during the test a solid radiolabeled meal is administered to the patient and a series of scintigraphic images is acquired: delayed GE is diagnosed if more than 60% of the meal is retained at 2 h or more than 10% of the meal is retained at 4 h [47]. GE scintigraphy, however, can be expensive and exposes patients to radioactivity. Moreover, GE is delayed by hyperglycemia, therefore, blood glucose levels should be controlled at the moment of the test. Ideally, glycemia should be lower than 200 mg/dL, if it is higher than 275 mg/dL the test cannot be performed or, in alternative, insulin should be administered to lower blood glucose levels [47].

An alternative, non-radioactive method for delayed GE documentation is the 13C-octanoic acid breath test [48], which has shown a strong correlation with GE scintigraphy in diabetic populations [46,49]. Hence, ^13^C-octanoic acid breath test represents a suitable alternative to investigate delayed GE in patients with DM in clinical practice.

Although many patients with DM have abnormal GE, few develop overt clinical symptoms, furthermore, part of symptomatic diabetics has little or no delay in GE. Differential diagnosis between gastroparesis and FD may be challenging, however, a delayed GE, the presence of vomiting and a lack of response to prokinetics are more suggestive of DG rather than of FD [10].

## 6. Clinical Management of Dyspepsia

The ACG/CAG clinical guidelines [5] provide indications on the diagnostic work-up which should be performed in patients with dyspeptic symptoms in addition to pharmacological therapies. According to guidelines, patients under the age of 60 should not undergo endoscopy to exclude malignancy, while, as previously mentioned, upper GI neoplasia should be excluded in elderly and in subjects with neoplastic risk factors [5]. The ACG/CAG clinical guidelines do not recommend the routine use of motility studies, which should only be performed in case of FD when gastroparesis is strongly suspected, as in patients with predominant symptoms of nausea and vomiting, who do not respond to empiric therapy. As discussed above, gastroparesis is diagnosed by documentation of delayed GE, investigated through GE scintigraphy or ^13^C-octanoic acid breath test, after exclusion of mechanical obstruction through radiologic or endoscopic examination [5].

Patients under the age of 60 should have a non-invasive test for *H. pylori* infection and they should be subsequently treated if the test is positive, while they should receive an empirical treatment with proton pump inhibitors (PPIs) if they are *H. pylori* negative or they are still symptomatic after bacterial eradication [5]. Even in the absence of gastric acid secretion abnormalities, PPIs showed to be effective in relieving FD symptoms and their efficacy was not related to concomitant GERD or *H. pylori* positivity [50].

Patients with dyspepsia not responding to PPIs and *H. pylori* eradication, can be offered a prokinetic therapy, despite the limited effectiveness data only available in non-diabetic dyspeptic patients [5]. However, only in dyspepsia related to DE, prokinetics have shown efficacy in improving gastric motility and reducing symptoms [30]. Prokinetics include serotonin-4 receptor agonists such as cisapride, mosapride and tandospirone citrate, which can be effective in relieving abdominal pain [51] and dopamine-2 receptor antagonists, like metoclopramide, which have shown efficacy similar to cisapride in improving GE and a better control of nausea, vomiting and early satiety [52]. However, metoclopramide is associated with important side effects, including hyperprolactinemia, closely related to gynecomastia and galactorrhea, and extrapyramidal symptoms, such as drug-induced parkinsonism and tardive dyskinesia [2].

Acotiamide is a prokinetic, currently approved for use in Japan and India for FD. It inhibits pre-sinaptic acetil-cholinesterase and antagonized presynaptic M1 and M2 receptors and it seems to relieve PDS symptoms, with a good tolerability [53,54].

An alternative to prokinetic drugs is represented by neuromodulators. In fact, triciclic antidepressant therapy (TCAs), such as amitriptyline, showed to relieve abdominal pain and improve the quality of life in patients with dyspepsia [55,56]. Data on serotonin reuptake inhibitors (SSRIs) and serotonin and norepinephrine reuptake inhibitors (SNRIs) are controversial as some studies showed an efficacy similar to TCAs [57], while other studies did not demonstrate any efficacy in symptom relief [53]. In the clinical setting the decision between prokinetics and TCAs should be made on a case-by-case basis [5].

As previously discussed, microbiota is receiving increasing attention in the context of FD and some authors have studied a potential effect of therapies targeting on gut microbiota, such as rifaximin [58] or supplementation with Lactobacillus strains [59], which act restoring the physiological microbiota. However, data about the indication to treat dyspeptic patients with probiotics remain scarce.

Finally, patients with FD not responding to drug therapy should be offered psychological therapies. Considering the role of psychological factors in the development of FD, in fact, these treatments may provide a significant symptom relief [10]. The quality of evidence about this approach is very low and the available studies are heterogeneous and do not suggest a specific psychological intervention [5].

Some authors have also proposed complementary and alternative treatments, such as herbal supplements, acupuncture and hypnosis [10], however, the available data are limited and more studies are needed to assess the efficacy of these therapies.

The above discussed therapeutic options for FD are summarized in Figure 1. Together with these pharmacological and non-pharmacological approaches, remains crucial the therapeutic role of lifestyle modifications, such as weight loss in obese patients, cessation of smoking, diet variations, NSAIDs avoidance [10]. These interventions represent the first step in FD treatment and should be associated with any other therapy.

## 7. Clinical Management of Gastroparesis

In regard to DG, a stepwise therapeutic approach is also recommended. There is a scarcity of data on appropriate dietary intervention with much of the data extrapolated from other conditions. Typically, dietary advice commences with a low-fat, low-fiber diet but may require liquid meals, enteral or parenteral nutritional support. Avoidance of drugs which delay GE, such as GLP-1 analogues and opioids is recommended [2]. As above described, higher glycemic levels are associated with delayed GE: an accurate glycemic control is therefore essential in the clinical management of DG. However, data on long-term improvement in terms of glycemic control are limited [38,42].

Prokinetic drugs, including metoclopramide and erythromycin represent the first-line therapy. Metoclopramide proved to significantly reduce symptoms of DG through both central antagonism on dopamine receptors and peripheral cholinergic effect [2], but its use is limited by the previously described side effects. Domperidone is a peripheral dopamine receptor antagonist with prokinetic effect, which showed to improve symptoms, with a positive effect sustained over time [2]. As domperidone does not cross the blood–brain barrier, the risk of hyperprolactinemia and extrapyramidal symptoms is significantly lower in comparison with metoclopramide. On the other side, it should be administered with caution in patients with impaired liver function or at increased risk of cardiac events (such as prolonged QT interval) and its co-administration with QT-prolonging drugs is contraindicated [2]. As regards erythromycin, early studies showed a significant reduction in the total symptom score after acute intravenous and chronic oral administration [60], however further investigations demonstrated that its long-term efficacy is often limited by development of tachyphylaxis [60,61]. Moreover, erythromycin is associated with potentially severe adverse events, such as QT prolongation and ventricular arrhythmia [60,62].

Prucalopride is a serotonin receptor agonist which is mainly used in the treatment of constipation. Recently, a randomized placebo-controlled study analyzed its efficacy in thirty-four patients with DG: prucalopride significantly reduced GE time and improved symptoms, evaluated through Gastroparesis Cardinal Symptom Index [63]. These data are promising, but still need to be confirmed in larger sample studies.

Additionally, antiemetics are useful for symptom control in patients with DG. Among them, aprepitant, a neurokinin-1 inhibitor, showed to increase gastric accommodation and reduce nausea and vomiting in DE, even if it had no effect on GE [62].

Agonists of ghrelin and 5-hydroxytryptophan receptors, which are still experimental, are giving promising results [2,37]. Relamorelin is a ghrelin agonist administered by subcutaneous injection, which showed to reduce symptoms and increase GE half-time in phase 2 trials [2,64]. Its main side effect was glycemic control impairment, with more frequent hyperglycemia episodes and higher HbA1c levels [2,64].

As above mentioned, a better comprehension of pathogenic mechanisms of DG could lead to new effective therapies. One of these possible research targets is represented by micro-RNAs. MiR-10b-5p regulates development and function of ICCs and pancreatic β cells through the KLF11-KIT pathway, in murine models, knockout of mir-10b in KIT+ cells led to DM and gastroparesis [65]. In these mice, injection of miR-10b-5p mimic or Klf11 small interfering RNAs were effective in improving glucose homeostasis and gastric motility [65], thus suggesting a potential therapeutic role of micro-RNAs.

As concerns the possible role of alternative medicine in DG, cannabinoids use has been suggested because of their positive effects on chemotherapy-induced nausea and vomiting. However, there are no studies investigating the use of synthetic or herbal cannabinoids in the symptomatic treatment of dyspepsia or gastroparesis [66]. Acupuncture, instead, showed promising results in improving gastric emptying in both murine models and human studies [66]. Its effects seem to be mediated by vagal activity, but other mechanisms could be involved [66]. In murine models, in fact, electroacupuncture was associated with reduced apoptosis and increased proliferation of ICCs [67].

In refractory DG, more invasive therapeutic approaches should be considered and evaluated on a case-by-case basis. These treatments include supplemental nutrition, preferably enteral, administered through a feeding jejunostomy, gastric electric stimulation, pyloromyotomy, and sleeve or total gastrectomy [2,38].

Hospital admission should be considered in case of gastroparesis associated to refractory vomiting, dehydration, electrolyte abnormalities and malnutrition [68]. Clinical management of hospitalized patients requires pharmacological control of symptoms, intravenous hydration, electrolyte correction, glucose control and nutritional support. Enteral nutrition should be preferred, even if in case of severe DG gastric feeding is often not tolerated, thus a nasoduodenal or nasojejunal tube placement can be necessary [68]. Although enteral nutrition should be the first choice, short-term parenteral nutrition may be needed for selected patients, when nasoenteric tube placement or feeding is not tolerated or is contraindicated [68].

Figure 2 shows a therapeutic algorithm for DG, which includes the above discussed treatment options.

## 8. Conclusions

FD and gastroparesis are characterized by a complex pathogenesis whose mechanisms remain unclear. This is even more true for patients with diabetes who often suffer from these disturbances. As a consequence, their management should be based initially on international guidelines and tailored to their individual needs. Well-designed studies are needed in this field.

## Figures and Tables

**Figure 1 jcm-10-01313-f001:**
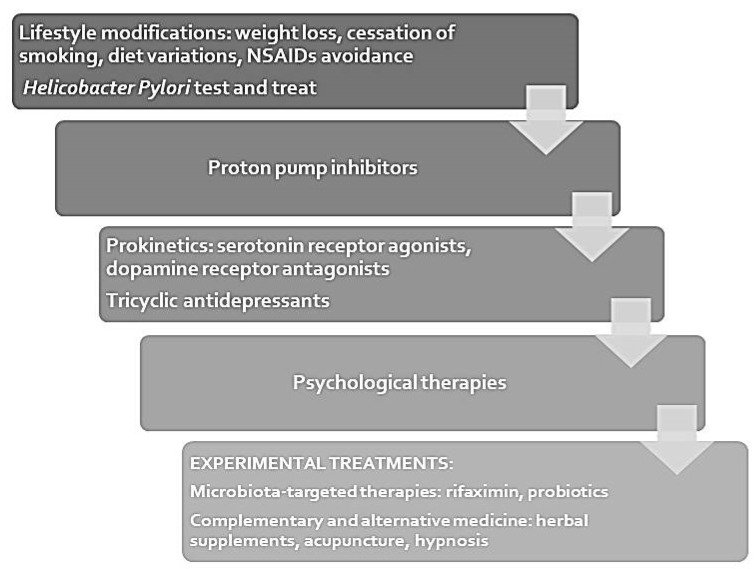
Treatment algorithm for functional dyspepsia.

**Figure 2 jcm-10-01313-f002:**
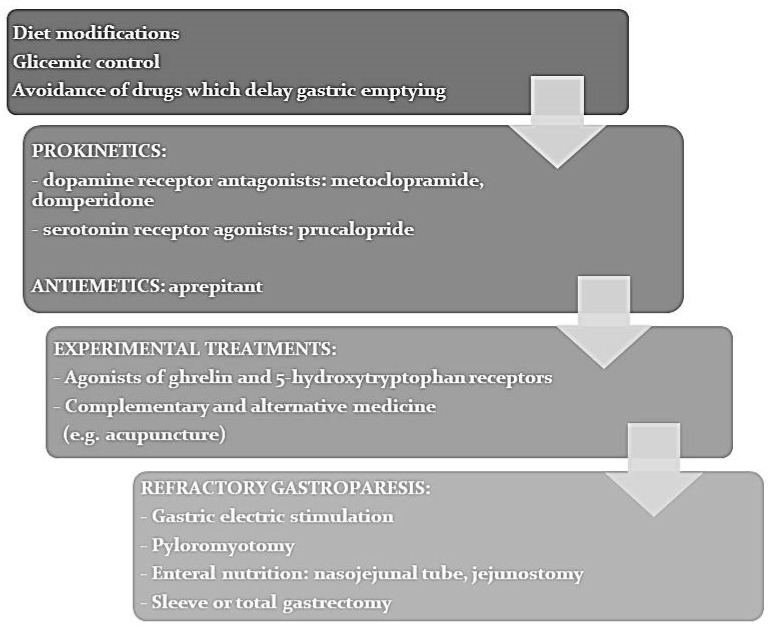
Treatment algorithm for diabetic gastroparesis.

**Table 1 jcm-10-01313-t001:** Rome IV diagnostic criteria of functional dyspepsia modified from [9].

**Functional Dyspepsia:**
1. One or more of the following: (a) postprandial fullness (b) early satiation (c) epigastric pain (d) epigastric burning AND 2. Exclusion of structural disease which can explain symptoms a. Must fulfill criteria for PDS (Postprandial Distress Syndrome) and/or EPS (Epigastric Pain Syndrome). b. Criteria fulfilled for the last 3 months with symptom onset at least 6 months before diagnosis.
**Postprandial Distress Syndrome (PDS):**
1. One or both of the following for at least 3 days per week and severe enough to impact on usual activities: (a) postprandial fullness (b) early satiation 2. No evidence of organic, systemic, or metabolic disease which can explain symptoms. Criteria fulfilled for the last 3 months with symptom onset at least 6 months before diagnosis. Supportive remarks: Possible co-existence of postprandial epigastric pain or burning, epigastric bloating, excessive belching. In case of vomiting, other disorders should be considered Possible association with heartburn Symptoms relieved by evacuation of feces or gas should not be ascribed to dyspepsia
**Epigastric Pain Syndrome(EPS):**
1. One or both of the following for at least 1 day per week and severe enough to impact on usual activities: (a) epigastric pain (b) epigastric burning 2. No evidence of organic, systemic, or metabolic disease which can explain symptoms Criteria fulfilled for the last 3 months with symptom onset at least 6 months before diagnosis Supportive remarks: Pain may be induced or relieved by ingestion of a meal or may occur while fasting Possible coexistence of postprandial epigastric bloating, belching, and nausea In case of persistent vomiting, other disorders should be considered Possible association with heartburn Pain cannot be defined as biliary pain Symptoms relieved by evacuation of feces or gas should not be ascribed to dyspepsia

**Table 2 jcm-10-01313-t002:** Differential diagnoses of dyspepsia.

ESOPHAGEAL DISEASES	Gastroesophageal reflux disease Eosinophilic esophagitis Achalasia Esophageal cancer
GASTRIC DISEASES	Peptic ulcer Erosive and non-erosive gastritis Helicobacter Pylori-related dyspepsia Gastroparesis Gastric cancer
DUODENAL DISEASES	Duodenal ulcer Duodenal cancer
PANCREATIC DISEASES	Acute pancreatitis Chronic pancreatitis Pancreatic cancer
HEPATOBILIARY DISEASES	Biliary lithiasis Cholangitis Cholangiocarcinoma
VASCULAR DISEASES	Superior mesenteric artery syndrome
SYSTEMIC DISEASES	Lymphoma Amyloidosis Connective tissue diseases (e.g., scleroderma)

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
