# Peer review of "Management of Dyspepsia and Gastroparesis in Patients with Diabetes. A Clinical Point of View in the Year 2021"

_jcm, 2021, doi:10.3390/jcm10061313_

Round 1
Reviewer 1 Report
Important topic to address.
Abstract:
P1.L17
Delayed GE should be performed to differentiate between FD and GP.
P1L20
Suggest adding to 1st line therapy prokinetics for GP dietary modifications as 1 line intervention and glucose control in diabetic patients. You should emphasize glucose control in diabetics.
Introduction
P1l30
Add present "with" upper GI symptoms
p1l36 delete "symptoms due to" -this is update on dyspepsia and GP in diabetic patients
p2l49
"instrumental investigations" poor term - use radiologic, endoscopic studies
p2l52
Work up may be more extensive to rule out pancreatic cancer or SMA syndrome, gallstones. You should add additional testing may be required US, CT, UGI depends on clinical scenario.
p3l64
Section on cancer and pancreatic disease may need modification suggest table to show list of conditions to consider DDx with dyspepsia -PUD, gallstones, cancer, SMA syndrome eosinophilic esophagitis...
p3 l81
it should be family Hx of cancer not "familiarity"
p 3 l 85
Sentence needs modification. you put acute and chronic pancreatitis together so weight loss is not symptom of acute pancreatitis.
p4 l155
What are muscular symptoms (describe ?arthralgia, myalgia)
p5 168
Add incidence/prevalence of GP in diabetes T1 and T2.
p5 l173
Recent mayo clinic data suggest gastroparesis correlates with glucose control not duration of diabetes! This makes sense if glucose is not controlled patients suffer more often from GP.
p5 l192
Is depression common in DM or only with diabetic GP? I believe patients with severe diabetes suffer from depression more often anyway.
P5 l199
Sentence with statement about Breath test with "year 1993" delete year 1993 do not add anything to this paper
Similar in sentence below p 5 204 with "published in 1983" delete year of publication does not add anything - add number of patients and method used for eval of GE.
p6 l219
Sentence needs modification. Gastroparesis is diagnosed by documentation of delayed GE after exclusion of mechanical obstruction (UGI or EGD).
Please add sentence with abnormal rates of GE at 2 and 4 hrs.
You should have a sentence on glucose levels and scintigraphy (GET) results.
Glucose levels should be measured during GET. if high the test result should be evaluated with caution
In management section you should have 2 sections on management of dyspepsia and separate on gastroparesis.
Suggest adding algorithm with therapeutic options and steps in management of both conditions.
Section on therapy is poorly written.
P.6 l270
1st line therapy in GP are prokinetics based on pathophysiology not antiemetics.
Antiemetics are often used for symptom control.
You need to discuss grelin agonists well studied in diabetic GP.
You need to put some discussion on domperidone. Discuss Metaclopromide side effect and limitations to its use.
You did not mention prucalopride or NK1 agonists.
Add in strategy glucose control.
Mention a role of marijuana if any?
p7.l1
Old metaanalysis on erythromycin.
Most studies showed that oral erythromycin performs poorly as long term Rx
Erythromycin IV works ok in combination with IV metaclopromide.
Consider reviewing recent paper by Vikram Rangan on gastroparesis in hospital setting in NCP 2020
Reviewer 2 Report
This is an interesting review. The authors have summarized the common pathophysiological mechanisms and current treatment options for functional dyspepsia and gastroparesis. However, there are some concerns listed below.
- The authors states that “Delayed gastric emptying may be documented through radioactive or non-radioactive methods and plays a key role in differentiating DG from dyspepsia?” Is this true? Authors should provide greater detail regarding the significant overlap of dyspepsia (FD) and gastroparesis and cite recent articles published in the Gastroenterology journal (Ref 1). This article suggested that FD and gastroparesis be grouped under one common clinicopathological spectrum of gastric neuromuscular dysfunction, further, they concluded that FD and gastroparesis cannot be distinguishable by clinical and pathological features or by assessment of gastric emptying. Thus, future studies are warranted to explore more pathologies underlying these conditions which might predict and correlate with symptoms better rather than currently just performing gastric emptying tests which do not correlate with the symptoms and degree of severity.
- The authors should discuss the common pathologies of FD and gastroparesis such as loss of interstitial cells of Cajal (ICCs) and reduction of CD206 expressing macrophage population in full-thickness stomach biopsies. For the future directions, also how these pathologies might be fixed through novel therapeutic approaches for better treatment and management with patients with FD and gastroparesis. Because relieving symptoms through prescribing prokinetics is not enough. Thus, there is an overwhelming demand to elucidate effective treatment regimens for FD and gastroparesis, targeting the underlying pathophysiology rather than only symptomatic treatment.
- Cite another article recently published in Gastroenterology (Ref 2). This study showed the fundamental role of microRNAs in the pathogenesis of diabetes and gastrointestinal dysmotility and demonstrated deficiency of miR-10b in pancreatic β cells and gastrointestinal ICCs in mice concurrently triggered diabetes and GI dysmotility, respectively. This study suggested miR-10b-5p could be clinically beneficial because it has profound and prolonged effects in lowering blood glucose and improving GI motility including gastric emptying in diabetic mice when compared to anti-diabetic and prokinetic medications.
- Authors should also discuss better treatment strategies for FD patients with EPS and PDS overlap as there is a significant clinical overlap between the EPS and PDS in the subgroups of FD. Further, the PDS subgroup is more likely to overlap with gastroparesis, and the EPS subgroup overlap with H. pylori-associated epigastric pain predominant.
- The authors should provide two tables for treatment approaches, it would be a good addition to the manuscript: (1) Treatment options for common pathologies in FD and DG such as gut microbial dysbiosis, visceral hypersensitivity, low-grade-inflammation) and (2) Treatment options based on the class such as prokinetics, antibiotics, antiemetics, and pain modulators.
- Helicobacter pylori (H. pylori) should be italicized throughout the manuscript.
References:
- Pasricha PJ, Grover M, Yates KP, Abell TL, Bernard CE, Koch KL, McCallum RW, Sarosiek I, Kuo B, Bulat R, Chen J, Shulman R, Lee L, Tonascia J, Miriel LA, Hamilton F, Farrugia G, Parkman HP; NIDDK/NIH GpCRC consortium. Functional dyspepsia and gastroparesis in tertiary care are interchangeable syndromes with common clinical and pathological features. Gastroenterology. 2021 Feb 3: S0016-5085(21) 00337-1.
- Singh R, Ha SE, Wei L, Jin B, Zogg H, Poudrier SM, Jorgensen BG, Park C, Ronkon CF, Bartlett A, Cho S, Morales A, Chung YH, Lee MY, Park JK, Gottfried-Blackmore A, Nguyen L, Sanders KM, Ro S. miR-10b-5p Rescues Diabetes and Gastrointestinal Dysmotility. Gastroenterology. 2021 Jan 7: S0016-5085(21)00001-9.
Round 2
Reviewer 1 Report
The manuscript covers the topics of dyspepsia and gastroparesis in very detailed way. I would like to congratulate the authors for an important contribution.